# Neural Hybrid Automata: Learning Dynamics with Multiple Modes and Stochastic Transitions

**Michael Poli**[*]
NAVER AI Lab, `DiffEqML`
poli@stanford.edu

**Stefano Massaroli**[*]
University of Tokyo, `DiffEqML`
massaroli@robot.t.u-tokyo.ac.jp

**Luca Scimeca**
NAVER AI Lab

**Seong Joon Oh**
NAVER AI Lab

**Sanghyuk Chun**
NAVER AI Lab

**Atsushi Yamashita**
University of Tokyo

**Hajime Asama**
University of Tokyo

**Jinkyoo Park**
KAIST

**Animesh Garg**
University of Toronto

## Abstract

Effective control and prediction of dynamical systems require appropriate handling of continuous–time and discrete, event–triggered processes. Stochastic hybrid systems (SHSs), common across engineering domains, provide a formalism for dynamical systems subject to discrete, possibly stochastic, state jumps and multi–modal continuous–time flows. Despite the versatility and importance of SHSs across applications, a general procedure for the explicit learning of both discrete events and multi–mode continuous dynamics remains an open problem. This work introduces *Neural Hybrid Automata* (NHAs), a recipe for learning SHS dynamics without *a priori* knowledge on the number of modes and inter-modal transition dynamics. NHAs provide a systematic inference method based on normalizing flows, neural differential equations and self–supervision. We showcase NHAs on several tasks, including mode recovery and flow learning in systems with stochastic transitions, and end–to–end learning of hierarchical robot controllers.

## 1 Introduction

Behaviors emerging from the interaction of continuous and discrete–time dynamics in the presence of uncertainty are described through the language of *stochastic hybrid systems* (SHSs). Such discrete events can bring along abrupt changes in the state, and in complex *multi–mode* systems, may also cause a *switch* between system *modes*, and corresponding underlying continuous dynamics [1]. Communication networks [2], [3], where changes in communication protocol can happen at certain levels of traffic congestion, and biological systems [4]–[6] are example domains where the SHS modeling paradigm has proven fruitful.

Data–driven identification and learning of hybrid systems are known to be challenging due to the entanglement of continuous flows and discrete events [7]; finding a generally applicable technique remains an open problem, particularly in the common scenarios where no *a priori* knowledge on the number and type of *system modes* is given. The aim of this work is to apply continuous neural models [8]–[10] to the learning SHSs. We introduce a compact descriptive language for this task, decomposing the system into a set of core primitives. Prior work is integrated into the framework, highlighting in the process limiting assumptions and areas of further improvement.

---

[*]Equal contribution. Author order was decided by flipping a coin.

35th Conference on Neural Information Processing Systems (NeurIPS 2021).

To address the shortcomings of existing techniques, we introduce *Neural Hybrid Automata* (NHA) as a general procedure designed to enable learning and simulation of SHSs from data. NHAs are comprised of three components: a dynamics module, taking the form of an *neural differential equation* (NDE) [8]–[10] capable of approximating a different vector field for each mode, a discrete latent state tracking the internal mode of the target system, and an event module determining the time to next event. In particular, our approach does not require prior knowledge on the number of modes. The synergy among NHA components ensures a broader range of applicability compared to previous attempts, which in example do not directly tackle multi–mode hybrid systems [11]–[14]. NHAs are shown to enable mode recovery and learning of systems with stochastic transitions, with additional applications in end–to–end learning of hierarchical robot controllers.

## 2 Background

We introduce required background on the formalism of *stochastic hybrid systems* (SHSs) and event handling for their numerical simulation. We then provide further contextualization on previous approaches, introducing in the process a unified language for SHS learning tasks.

### 2.1 Stochastic Hybrid Systems

A *stochastic hybrid system* (SHS) [2], [15] is a right-continuous stochastic process $X_t$ taking values in $\mathbb{X} \subseteq \mathbb{R}^{n_x}$ with a *latent mode* process $Z_t$ conditioning the dynamics of $X_t$, where $t \geq 0$. $Z_t$ is another right-continuous stochastic process that takes values in a finite set $\mathbb{M}$ of size $m$. In this context, the set $\mathbb{M}$ contains *identifiers* of internal system *modes*. An *event* is defined as either a mode switch or a state discontinuity (a *jump* in $X_t$), which can in some cases occur simultaneously. We refer to times at which events $z \to z'$ occur as random variables $t_k \in \mathcal{T}$, with associated *intensity functions* [16]

$$\lambda_{z \to z'}(t | \mathcal{H}_t) \geq 0.$$

where $\mathcal{H}_t := \{t_k \in \mathcal{T} : t_k < t\}$ is the *history* of event times. Intensity, as defined in the classical *temporal point process* (TPP) sense, can be interpreted as the expected number of events $z \to z'$ within the time interval $[t, t + \mathrm{d}t]$. The dynamics for $X_t$ when $Z_t = z$ is defined by

$$\text{flow dynamics}: \quad \dot{x}_t = F_z(t, x_t). \qquad (z, t, x_t) \in \mathbb{M} \times \mathbb{T} \times \mathbb{X}$$

When a jump event $z \to z'$ is triggered, $X_t$ can instantaneously jump according to

$$\text{jump dynamics}: \quad x_t^+ = \psi_{z \to z'}(t, x_t). \qquad (z, z', t, x_t) \in \mathbb{M}^2 \times \mathbb{T} \times \mathbb{X}$$

Jump maps $\psi_{z \to z'}$ and intensities $\lambda_{z \to z'}$ describe the behavior during events $z \to z'$.

### 2.2 Event Handling for Hybrid Systems

Following [17], to enable forward simulation of SHSs, a convenient mathematical representation of an *event* is a function $g : \mathbb{T} \times \mathbb{X} \to \mathbb{R}$ which nullifies only at any *event time* $t^*$, thus providing the differential equation integration algorithm with a *termination* or *restart* condition i.e.

$$t^* \text{ is an event} \iff g(t^*, x_{t^*}) = 0. \qquad (2.1)$$

The particular form of $g$ induces a jump set on $\mathbb{D} \subset \mathbb{T} \times \mathbb{X}$, $\mathbb{D} := \{t, x : g(t, x_t) = 0\}$, and determines transitions from roots of $g$ to regions of the state–space $\mathbb{X}$ where $g \neq 0$. Notably, this construct enables utilization of root finding methods [18] in a neighborhood of $t^*$ to accurately zero in on the event time.

The same simulation technique can be extended to the *many* jump sets case typical of multi–mode systems, by equipping the condition function with identifiers $z$, $z'$ ($g_{z \to z'}$) which induces jump sets $\mathbb{D}_{z \to z'}$.

**Simulating stochastic events** While the event function approach appears to be limited to the deterministic setting, it also subsumes stochastic events whose aleatoric uncertainty is encoded by an intensity $\lambda(t | \mathcal{H}_t)$ [1], [13]. Without loss of generality let us consider a single intensity function which is henceforth denoted as $\lambda_t^* := \lambda(t | \mathcal{H}_t)$. Recalling that the *cumulative distribution function*

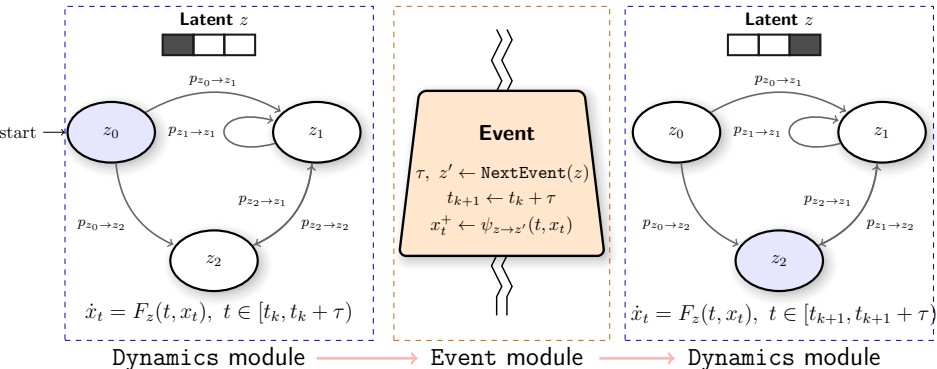

**Figure 1:** Schematic of a Neural Hybrid Automata (NHA). The mode–conditioned Neural ODE $F_z$ drives the system forward until an event time $t_{k+1}$ determined by a previous call to the event module. Then, the event module determines time to next event and corresponding mode target $z'$ through sampling from normalizing flow $p_{z \to z'}$ approximating densities of interevent times. A jump function is then applied to the state $x$, and simulation continues with flow $F_{z'}$.

(CDF) of inter–event times is $1 - \exp\left\{ -\int_{t_0}^{t^*} \lambda_t^* \mathrm{d}t \right\}$, standard *inverse transform sampling* [1], [19] yields

$$t^* : 0 = s - \log \int_{t_0}^{t^*} \lambda_t^* \mathrm{d}t, \;\; s \sim \texttt{Uniform}(0, 1) \tag{2.2}$$

as a special case of (2.1). Approaches developed for learning TPPs, including in the context of Neural ODEs [8], [11], [20], introduce a parametric formulation for the intensity $\lambda_\theta^*$ and optimize via direct TPP likelihood objectives. The integral is in general intractable, thus these methods require either a numerical approximation or the augmentation of additional states to compute it alongside the ODE.

## 2.3 Core primitives for SHS learning

At minimum, a learning model for SHSs necessitates several modules, each mirroring an element of the formulation in 2.1. More specifically:

 $i.$ Dynamics module, to approximate continuous dynamics $F_z$ conditioned on mode $z$.

 $ii.$ Discrete latent selector, to identify at each event time the latent mode $z$ of the system.

 $iii.$ Event module, to determine *when* events happen, and how state $x$ and latent state $z$ are updated after the transition.

Prior work considers specific instantiations of SHSs, leading to simplifying choices for each module defined above. In example, switching systems without jumps, where $iii.$ does not require jumps [21], single mode systems, where $ii.$ is not required and $iii.$ does not need to determine latent mode transitions [11], [13], [14], systems with known dynamics, where $i.$ is not trained [13], or systems with only deterministic events [7]. We note some of these works suffer from more than a single of these limitations, including additional ones consequence of specific model choices. Direct parametrization of the intensity, while a reasonable choice for single mode systems, requires state augmentation scaling in the worst case as $\binom{m}{2}$ [22] for a SHS with $m$ modes. More importantly, training parameters $\theta$ for such direct approaches is affected by the accuracy of the numerical method employed for the solution of the integral in (2.2).

In this work, we introduce a modelling framework SHSs that does not rely on any of the simplifying assumption on $i.$, $ii.$ and $iii.$ outlined above.

# 3 Neural Hybrid Automata

We introduce *Neural Hybrid Automata* (NHA), a model for learning of SHSs. A NHA is comprised of a dynamics module, a discrete latent state and an event predictor. A general overview of a NHA is depicted in Figure 1. We start with a description of each module and their interconnections, followed by a step–by–step procedure for NHA training.

**Event module** Intensity–free parametrizations for stochastic mode transitions allows NHAs to sample next event times without solving integrals $\int \lambda_t^* \mathrm{d}t$ for all target modes $z'$ reachable from current $z$. From event $k$ at time $t_k$, NHAs determine next event times $t_{k+1}$ through a conditional normalizing flow modeling, for each possible pair of $(z, z')$, the density of corresponding *inter–event* times $\tau_{z \to z'}^k = t_{k+1} - t_k, t_k \in \mathcal{T}_{z \to z'}$. Let the intensity be a simple *timer* i.e $\dot{\lambda} = 1$. Further, let $p_{z \to z'}$ be the parametrized conditional density obtained by the normalizing flow and let $T(z, z', t_k)$ be a collection of conditional samples from $p_{z \to z'}$ (one for each pair $z, z'$), i.e.

$$T(z, z', t_k) = \left\{ \tau_{z \to z'}^k \sim p_{z \to z'}(\tau | \mathcal{H}_{t_k}) \right\}_{z, z' \in \mathbb{Z}}.$$

Using (2.2), we can thus sample an event given the current mode $z$ and the previous event time $t_k$ as:

$$t_{k+1} = t_k + \min_{z' \in \mathbb{Z}} T(z, z', t_k). \tag{3.1}$$

Note that the next mode $z'$ after the event is simultaneously obtained as $z' = \arg \min T(z, z', t_k)$. Sampling strategy (3.1), differently from (2.2), relies on the normalizing flow to explicitly model the density rather than defining it implicitly through $\int \lambda_t^* \mathrm{d}t$. When event time $t_k$ is reached, a parametrized jump map conditioned on $(z, z')$ is applied to the state $x^+ = \psi_{z \to z'}(t_k, x)$. Normalizing flow $p_{z \to z'}$ and jump map $\psi_{z \to z'}$ together define the full event module of an NHA.

It should be noted that (3.1) always samples the *quickest–to–occur* event from the normalizing flow, which implies that no other event occurs between $t_k$ and $t_{k+1}$. While the history can be compressed into a fixed–length vector following [23] through application of sequence models e,g. RNNs, we note that for hybrid systems equipped with deterministic events, providing $(x_{t_k}, t_k)$ as conditioning inputs for $p_{z \to z'}(\tau)$ is sufficient since ODE solutions with deterministic transitions are uniquely determined by the initial condition. Finally, deterministic events are a special case of stochastic events [2] that can be well–represented with a Dirac $\delta$ function, of which the normalizing flow learns a smooth approximation with continuous support.

**Dynamics module** To enable approximation of different mode–dependent vector fields, we parametrize the *flow map* $F_z(t, x_t)$ of a SHS as a *data–controlled neural ordinary differential equation* (Neural ODE) [9] with parameters $\omega$, driven between each pair of event times $t_k, t_{k+1}$ by discrete latent mode $z$

$$\dot{x}_t = F_z(t, x_t, \omega) \qquad t \in [t_k, t_{k+1}) \tag{3.2}$$

Finiteness of admissible values in the latent mode state i.e. $m$ ensures $F$ is capable of approximating a finite number of different vector fields, one for each mode. In particular, we consider one–hot representations for latents $z \in \mathbb{R}^m$. In batched data settings, (3.2) can be integrated in parallel across $n_b$ batches of initial conditions $x_{t_k} \in \mathbb{R}^{n_b \times n_x}$ with different modes, provided the latent is also batched $z \in \mathbb{R}^{n_b \times m}$.

The combination of a given dynamics and event module, applied in turn as depicted in Figure 1, enables simulation of trajectories of a SHS. We now describe their training procedure.

# 4 Neural Hybrid Automata Module Training

Here, we detail the training procedure for each NHA component. Our only assumption is availability to a *trajectory segmentation* routine tasked with separating the trajectories, or flows, into a collection of *subtrajectories* $X_i$ of potentially of different length, each produced by the system in a different mode. The routine can be as simple as detection of discontinuities in the solution by inspecting *finite–differences* of observations across timestamps [24], or involve additional steps such as change–point detection [25]. Providing exact event times to NHAs is not required; the segmentation routine need only partition the full dataset in $n$ disjoint sets $X_i$ s.t. $\bigcup_i X_i = X$ and $\bigcap_i X_i = \emptyset$. In addition, no knowledge of the number of modes, or topology of transitions between modes is made available to NHAs, as these are rarely available in practice.

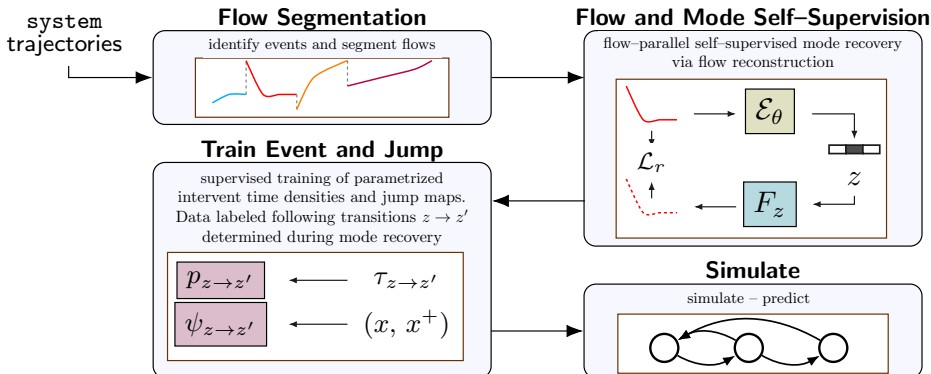

**Figure 2:** NHA training blueprint. Segmenting the trajectories enables self–supervised mode recovery via trajectory reconstruction. The recovered mode labels are then used for NHA event module supervised training.

**Self–supervised mode recovery** The first stage of learning an NHA is designed to approximate the continuous dynamics under each SHS mode while simultaneously identifying modes $z$. We achieve this by framing subtrajectory reconstruction as a *pretext* task for mode recovery, via a reconstruction objective $\mathcal{L}_r = \frac{1}{n} \sum_{i=0}^{n} \ell_r(X_i, \hat{X}_i)$, being $\hat{X}_i$ subtrajectories reconstructed by the flow decoder $F_z$ via the model

$$
\begin{aligned}
z &\sim \mathcal{E}(X, \theta) & t &= t_k \\
\dot{x} &= F_z(t, x_t, \omega) & t &\in [t_k, t_{k+1}).
\end{aligned}
\tag{4.1}
$$

Here, a latent *encoder* $\mathcal{E}$ with parameters $\theta$ is tasked with extracting a latent mode state $z \in \mathbb{M}$ to steer the decoder $F_z$ towards a more accurate reconstruction. Representation limitations of Neural ODEs [9], [26] ensure that to fit the above objective the the encoder $\mathcal{E}$ has to cluster the trajectories to enable the data–control decoder to represent different vector fields for each system mode. Finiteness of admissible values in the latent state is enforced by defining $z$ as one–hot encoded sample from a parametrized categorical distribution. Backpropagating through the sampling procedure is performed via straight–through gradients [27]. System (4.1) can be regarded as an ODE trajectory autoencoder with a categorical bottleneck.

Finally, we note that trajectory segmentation serves multiple purposes during NHA training. Forward integration is significantly sped up since the ODE solves can now be parallelized across subtrajectories $X_i$ as independent samples of a batch of data, avoiding a sequential solve on full SHS trajectories. The speedups can be dramatic for multi–mode SHSs[2], the focus of this work, where data trajectories may need to be longer to sufficiently *explore* different modes.

**Event and jump supervision** In addition to the learning of mode dynamics, self–supervised mode recovery objectives provides direct supervision for normalizing flows $p_{z \to z'}$ and jump maps $\psi_{z \to z'}$. More specifically, we collect times $\tau_{z \to z'}^k$ and jump state pairs $(x, x^+)$ for each pair of modes $(z, z')$ corresponding to a transition between pairs of subtrajectories clustered as $z$ (first) and $z'$ (second) by the encoder $\mathcal{E}$. We then train the jump maps $\psi_{z \to z'}$ to approximate $x \mapsto x^+$, and the mode conditional normalizing flow to approximate the density $p_{z \to z'}(\tau)$.

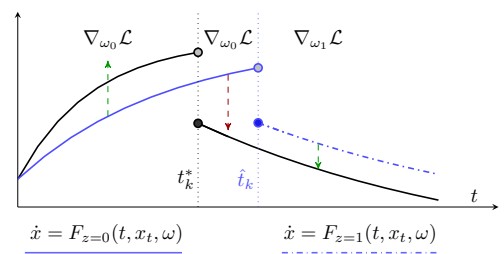

**Figure 3:** Conflicting gradients in an idealized 2–mode hybrid system due to overestimation of event time.

**Gradient pathologies in joint learning of flows and events** When attempting simultaneous learning on the full trajectory, the parameters of flow $F_z$ can be subjected to wrong gradients from reconstruction objectives, arising from overreliance or underreliance of the flow model on certain

---

[2]While speedups are dependent on full trajectory and average subtrajectory lengths, in our experiment we observe at least an order of magnitude (more than 20x) in wall–clock speedups for a single training iteration.

modes. This phenomenon can bias training, and provides strong motivation behind our segmentation–first approach, since each subtrajectory $X_i$ is associated only to a single mode[3].

A visualization is provided in Figure 3, through an idealized learning task of a two–mode system. Overreliance of the flow model on mode $z = 0$, due to overestimation of event time $t_k$, leads to a decomposition of gradients $\nabla_{w_0}\mathcal{L}_r$; in green, gradients pushing the trajectory closer to the solution, in red, incorrect gradients pushing the mode 0 trajectory further away from the ground–truth and closer to a solution belonging to a different mode. Appendix A further develops theoretical considerations on the nature of these gradients.

# 5 Results and Discussion

We evaluate *Neural Hybrid Automata* (NHA) through extensive experiments, with a focus on investigating the performance and robustness of each NHA module. A summary of experiments, objectives and ablations is provided here for clarity:

- **Reno TCP**: we carry out a quantitative evaluation on quality of learned flows (mean squared error) and quality of mode clusters recovered during self–supervision (v–measure). We also verify the robustness of NHAs to overclustering and amount of data required for event module training.

- **Mode mixing in switching systems**: we highlight and varify robustness against *mode mixing*, a phenomenon occurring during learning of multi–mode systems through alternative *soft* parametrization of latent $z$, such as through `softmax` instead of `categorical` samples.

- **Behavioral control of wheeled robots**: NHAs enable task–based behavioral control. We investigate a point–to–point navigation task where a higher level *reinforcement learning* (RL) planner determines mode switching for a lower–level optimal controller.

## 5.1 System with Stochastic Transitions

We apply NHAs to a dataset of internal state trajectories of a network transmission controller (TCP), the Reno TCP scheme [2]. The system has two states, five stochastic transitions and three modes as shown through an automata representation in Figure 4. Here, we qualitatively validate the performance of dynamics and event modules of NHAs. We simulate 40 trajectories of the system, each 200 seconds long, and segment them. No a priori knowledge on the mode of each subtrajectory is provided to the model. We perform self–supervised mode recovery to train $F_z$ and $\mathcal{E}$, in the process labeling the subtrajectories, then train event module normalizing flows and jump functions with the mode labels obtained. Training and evaluation are performed using a 5–fold cross–validation strategy, with a final test fold of 15. More details on the system, architectures and data generation are reported in the Appendix B.

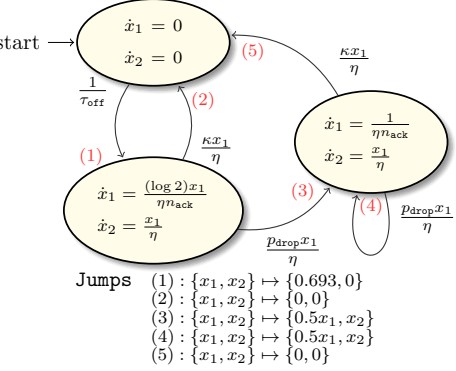

Jumps  $(1) : \{x_1, x_2\} \mapsto \{0.693, 0\}$
$(2) : \{x_1, x_2\} \mapsto \{0, 0\}$
$(3) : \{x_1, x_2\} \mapsto \{0.5x_1, x_2\}$
$(4) : \{x_1, x_2\} \mapsto \{0.5x_1, x_2\}$
$(5) : \{x_1, x_2\} \mapsto \{0, 0\}$

**Figure 4:** Automata representation of TCP Reno, where $\eta$, $p_{\text{drop}}$, $\kappa > 0$ and we set $n_{ack} = 2$. On each edge is the corresponding intensity $\lambda_{z \to z'}$.

**Mode recovery results** First, we perform self–supervised mode recovery and verify (i) whether the mode conditioned NHA decoder $F_z$ offers test–time TCP trajectory reconstruction of equal or better quality than other Neural ODE variants, and (ii) quality of the mode label clusters assigned by the NHA encoder and robustness to different mumber of latent modes $m$. We measure (i) via test *mean squared error* (MSE) on reconstructed trajectories, and (ii) via *v–measure* [28], a metric taking values in $[0, 1]$, computed as the harmonic mean between cluster *completeness* and *homogeneity*. A v–measure of 1 indicates perfect clustering. As baselines, we collect for (i) the performance of 3 *Neural ODE* (NODE) variants, a zero–augmented NODE, a *data–controlled NODE* (DC–NODE) [9] where the latent $z$ is the output of a multi–layer encoder, and a Latent NODE where $z$ is sampled

---

[3]Due to inaccuracies or noise in the segmentation algorithm, these partitions might not be perfectly separated into different modes. We experimentally investigate these effects on NHA training in Appendix B.

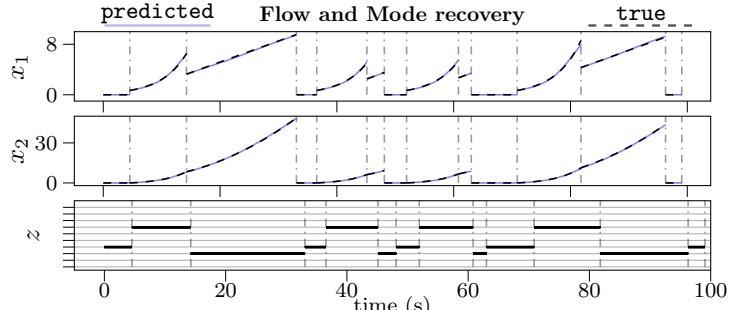

| Model | Test RMSE |
|---|---|
| NODE | 3.57 |
| DC–NODE | 1.46 |
| Latent NODE | 1.55 |
| NHA–3 | 1.66 |
| NHA–5 | 1.21 |
| NHA–10 | **0.56** |

**Figure 5:** **[Left]** Reconstruction of system trajectories through NHA vector field decoders $F_z$ and corresponding modes $z$ encoded by $\mathcal{E}$ for Reno `TCP`. Although the encoder shown is initialized with more modes (10) than there are in the underlying system (3), mode clustering is sparse and accurate. **[Right]** Flow reconstruction test RMSE for different classes of decoders, selected via cross-validation on 10 runs. NHA decoders can reconstruct the flows as well as other NODE baselines, with the added benefits of being able to recover mode labels during training. NHA$-m$ indicates NHA decoders initialized with $m$ modes.

via reparametrization of a Normal [8]. We also provide baseline performance of a series of popular clustering algorithms tasked to cluster the subtrajectories: k–means++ [29], hierarchical [30] and DBSCAN [31]. Figure 5 provides qualitative and quantitative results for stage (i). As established by Table 1, NHA mode recovery outperforms all baselines by a wide margin, with v–measure values close to 1. Surprisingly, we observe providing the NHA encoder with a larger number of latent modes than the 3 of the system improves clustering results and stabilizes training. Additional details on data pre–processing, metrics and baseline design and tuning are provided in Appendix B.

**Event module results** Next, we leverage the mode labels recovered as supervision for the event module of an NHA. In all cases, we train three–layer MLPs as jump maps and two–layer spline flows [32] as normalizing flows. Figure 6 visualizes the learned densities for each stochastic transition for the standard training regime of $n = 5$ trajectories. We also perform an ablative study on the quantity of data required to extract sufficient supervision signal for both components of the event module. The results are included in Table 2. We find that a single trajectory is sufficient, with relative performance gains quickly dropping off after $n = 3$.

| | v–measure ↑ | | |
|---|---|---|---|
| **Model** | $m = 3$ | $m = 5$ | $m = 10$ |
| k–means++ | $0.20 \pm 0.02$ | $0.24 \pm 0.02$ | $0.30 \pm 0.06$ |
| hierarchical | $0.23 \pm 0.01$ | $0.24 \pm 0.01$ | $0.31 \pm 0.06$ |
| DBSCAN | $0.66 \pm 0.02$ | $0.68 \pm 0.02$ | $0.69 \pm 0.01$ |
| NHA | $\mathbf{0.91 \pm 0.09}$ | $\mathbf{0.95 \pm 0.04}$ | $\mathbf{0.96 \pm 0.03}$ |

**Table 1:** Quality of recovered mode clusters from NHA self–supervised training and baseline clustering algorithms in the TCP task. Hyperparameter $m$ is the number of clusters provided to each algorithm. For DBSCAN, values of $m \in [3, 5, 10]$ map instead to its primary parameter $\epsilon \in [0.1, 0.5, 1]$ [31].

| Model | Metric | $n = 1$ | $n = 3$ | $n = 5$ | $n = 10$ |
|---|---|---|---|---|---|
| $p_{z \to z'}$ | NLL ↓ | 2.761 | 2.375 | 2.362 | 2.313 |
| $\psi_{z \to z'}$ | MSE $(10^{-3})$ ↓ | 1.435 | 0.018 | 0.009 | 0.003 |

**Table 2:** Quality of fit for event module components, normalizing flows $p$ and jump maps $\psi$. Training performed with supervising mode labels from $n$ trajectories of TCP. We report test MSE and *negative log–likelihood* (NLL) estimated from a base normalizing flow model trained on ground–truth data from $n = 500$.

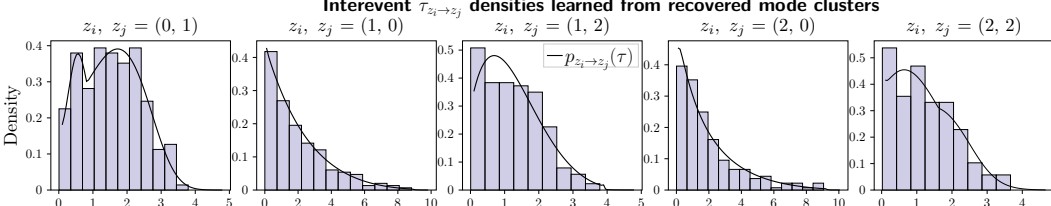

**Figure 6:** Learned densities (black) for intervent times $\hat{\tau}_{z_i \to z_j}$. The normalizing flows are trained on times recovered during the mode recovery stage, by clustering $\tau_{z_i \to z_j}$ according to the encoder mode labeling. The histogram depicts the ground–truth empirical distribution for each class of event.

## 5.2 Deterministic Switching System

We investigate *mode mixing* in a three–mode switching linear system (SLS) [13]. The deterministic nature of mode transitions, as well as the absence of state jumps enables direct training of NHAs on data–trajectories without prior segmentation. This allows us to perform an ablative study on *mode mixing*, a phenomenon arising from latent modes $z$ produced by the encoder $\mathcal{E}$ via *soft* alternatives to categorical samples, such as by using `softmax` activations.

**Mode mixing and overclustering**   We train NHAs on reconstruction of SLS trajectories. Each encoder is provided, at initialization, one additional latent mode over the three of the system. The conditioned flow $F_z$ is constructed with three–layer MLPs. Rather than segmenting the data, we repeat sampling for $z$ at each integration step. Figure 7 shows the state space switching boundaries and mode vector fields learned by an NHA and a baseline producing $z$ via `softmax` rather than as categorical samples. The additional freedom provided by softmax latents $z \in \mathbb{R}^4_+$, $\sum z_i = 1$ allows fitting the trajectories by nonlinearly mixing different vector fields to approximate a single one. Instead, categorical samples cannot mix the vector fields; this ensures that the learned clustering is either sparse as shown in the Fig.7, or latent values dedicated to the approximation of the same underlying mode dynamics are forced to learn the same vector field. Appendix B contains a visualization and analysis for this second case.

In general, categorical bottlenecks are effective when recovery of ground–truth system mode dynamics is a primary objective, particularly as it allows pruning of redundant modes as discussed in Appendix B. `Softmax` or other soft relaxations can be a viable choice if only black–box fitting of data is desired.

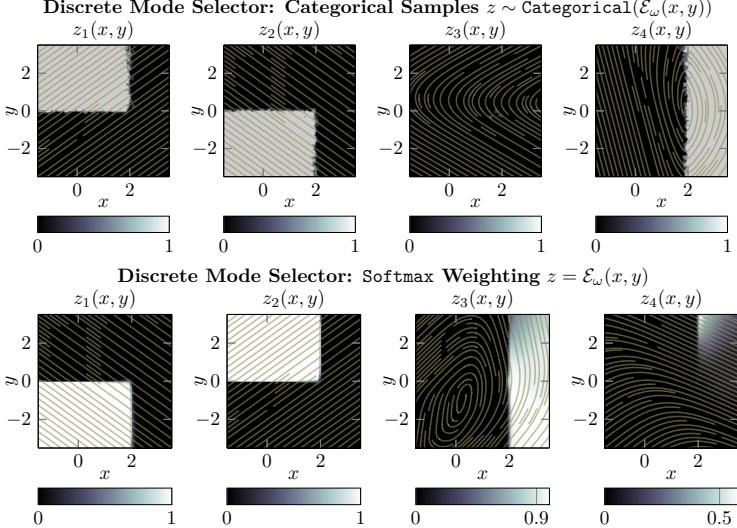

**Figure 7:** Reconstructed conditional vector fields $F_z$ and corresponding mode classification boundaries in the state–space of the LSS. In white, the region of the state space assigned to mode $z_k$. **[Above]** Categorical NHA encoder. **[Below]** Mode classification performed by "soft" encoder $\mathcal{E}$ capped with a `softmax` activation. Soft encoders mix vector fields to approximate trajectories, and are thus unable to recover the ground–truth dynamics for each mode.

## 5.3 End–To–End Learning of Hierarchical Switching Controllers for Dynamical Systems

Beyond SHS identification, the NHA framework enables learning of task–based hierarchical controllers comprising a low–level controller $u_z := u(t, x_t, z)$ dependent on the discrete mode $z$ provided by a higher–level policy $\pi$. Each NHA module is adapted as:

$$\text{dynamics module: } \dot{x}_t = F(t, x_t, u_z) \quad \text{low–level controlled system}$$
$$\text{event module: } z' \leftarrow \pi(t, x_t, z) \quad \text{high–level planner} \tag{5.1}$$

Within this context, latent state $z$ can be regarded as a system *set–point* (e.g. a desired value of state $x$) determined by the planning policy $\pi$ to achieve a certain task, which the low–level controller has then to carry out. Both $\pi$ and $u_z$ are parametrized by neural networks, and the training is done end–to–end.

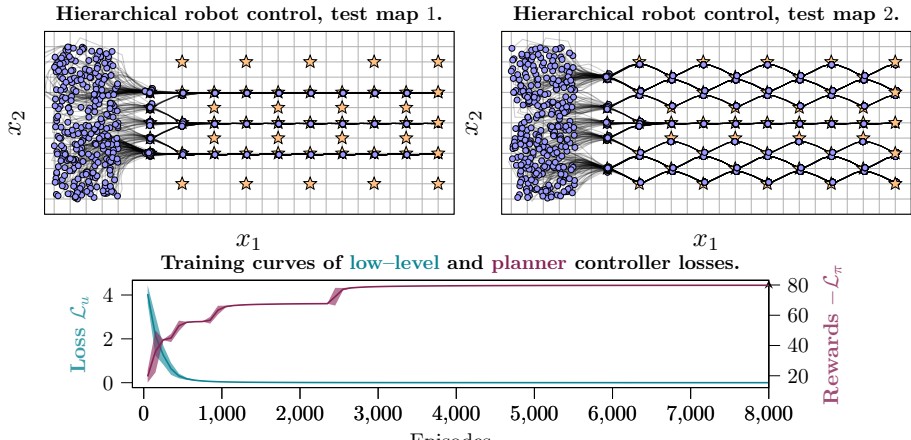

**Figure 8: [Above]**: Test–time learned navigation of swarms of differential drive robots. The robots are initialized at random locations and orientations. **[Below]**: Training loss curves of low–level controller $u$ and planner $\pi$.

The obtained hierarchical control scheme is sample efficient, since system dynamics available a priori are included in $F$. From the perspective of $\pi$, however, the dynamics are *disentangled* from the planning objective. Indeed, the higher level policy need only learn how to set and switch between objectives ($z \to z'$), and not how to control the system to reach them. For this reason we train the model using a loss function partitioned in two terms as $\mathcal{L} = \mathcal{L}_u + \mathcal{L}_\pi$.

**Results**   We consider learning controllers for navigation of two-wheeled differential drive robots [33]. The higher–level model–free policy $\pi$ is here trained via REINFORCE gradients [34], [35] to select the nearest resource target every $5$ seconds. We set $5$ different resources within a map, and train all robots to: one, select the nearest resource target; two, drive the wheels to reach the target. The training of both controllers is carried out concurrently, where target (or *mode*) selection is performed by $\pi$, whereas the behavioural controller $u_z$ has to reach the target chosen by $\pi$ via low–level steering control inputs. As shown in Figure 8, convergence of both control policies occurs after around $4000$ episodes of training. Figure 8 also visualizes the resulting navigation behavior at test–time on two new resource layouts, where we alternate between two different sets of targets.

## 5.4   Generalizable Insights and Empirical Observations

The task of learning SHSs involves several moving components. Ablative experiments have been performed to address specific questions on the robustness of NHAs. We detail heuristics that have been observed to improve performance, and report areas of further improvement.

• **Overclustering stabilizes training**   We empirically observe providing NHA with more latent modes than the system stabilizes training. We conjecture the additional choice allows the model to use different modes during exploration without always conflicting with other already "assigned" modes, phenomenon which is more frequent, in example, when the number of modes exactly matches that of the system. In these cases, dropout in the encoder $\mathcal{E}$ appears to improve performance.

• **Decoder expressivity limitations improve mode recovery**   Effective mode clustering via the trajectory autoencoder introduced in Section 4 relies on representational limitations of Neural ODEs [9]. Ensuring a sufficient state–space density of subtrajectories improves quality of mode clusters.

• **Noisy segmentation of trajectories**   We investigate, for the TCP experiment, robustness of mode recovery to incorrect segmentation (Appendix B) and number of NHA latent modes $m$ (Table 1). Extending NHAs to include a finetuning step for trajectory segmentation, in example leveraging ideas from [13] might improve robustness of the segmentation routine and thus the overall approach.

# 6 Related Work

**Hybrid system identification and Markov models**  Hybrid system identification is a relatively recent development in dynamical system theory [7]. A majority of existing literature focuses on (linear) *piecewise affine systems* (PWA) [36], [37]. [38] proposes a clustered symbolic regression algorithm for learning input–output maps rather than dynamics. Existing approaches involving continuous optimization [21] do not consider event stochasticity and mode recovery. Identification of SHSs is an even smaller field, with limited success outside specific cases [15].

**Continuous–depth and contact models**  Neural differential equations and continuous–depth models, initially concerned with unimodal systems [8], [9], have seen preliminary application to the learning of temporal point processes [11], [39]. Although some of these works tackle stochastic events and marked point processes, multimodality and explicit learning of the flows is not considered. [12] examine *interventions* as events, and develop a continuous architecture for modeling the lasting effect of a given intervention on the dynamics. Differentiable contact approaches [14], [40] introduce physics–compatible models designed to recover deterministic hybrid dynamics of mechanical systems from data. [13] develops, through implicit differentiation, a method for direct optimization of event times. Although Neural Event ODEs do not directly address multimodality, a potential synergy between the approach of [13] and NHAs could preserve the advantage of our flow–parallel mode recovery, namely integration speed and sidestepping of gradient pathologies outlined in Section IV. Section 2.3 provides a summary of limitations for these existing methods.

# 7 Conclusion

Hybrid systems represent a versatile and general class of systems, with applications across engineering disciplines [15]. In this work, we investigate challenges related to the learning of SHSs from limited data and introduce *Neural Hybrid Automata* (NHA), a step–by–step method leveraging neural differential equations, density estimation and self–supervised mode recovery. NHAs are shown to be effective in various settings, including flow and event learning in systems with stochastic transitions.

## Funding Statement

This work was financially supported by NAVER AI Lab, KAIST and University of Tokyo. All experiments were run on GPUs provided by KAIST and University of Tokyo.

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
