# Neural Hybrid Automata
## *Supplementary Material*

## Table of Contents

## A   Additional Discussion and Theory

### A.1   Neural Hybrid Automata: Modules and Hyperparameters

We provide a notation and summary table for *Neural Hybrid Automata* (NHA). The table serves as a quick reference for the core concepts introduced in the main text.

1. Dynamics: tasked with approximating continuous dynamics of each mode by conditioning a Neural ODE on mode $z$.

2. Mode Encoder: only used during self–supervised mode recovery. Labels every subjtrajectory $X_i$ with a mode $z$ to ensure mode–conditioned decoder $F_z$ can reconstruct it despite Neural ODE representation limitations (uniqueness of solutions given an initial condition).

3. Event Module: determines during simulation (i) when events happen, and what types of events i.e. mode transitions $(z \rightarrow z')$ through $p_{z \rightarrow z'}$, (ii) what happens during such events i.e. jumps on the state via $\psi_{z \rightarrow z'}$. Normalizing flow $p_{z \rightarrow z'}$ is trained to approximate densities $p(\tau_{z \rightarrow z'}|\mathcal{H})$.

The only NHA hyperparameter beyond module architectural choices is $m$, or number of latent modes provided to the model at initialization. Performance effects of changing $m$ have been explored in Section 5.2 and Appendix B.2. Appendix B.2 further provides analyzes potential techniques to prune additional modes.

Dynamics:    $\dot{x} = F_z(t, x_t, \omega)$   $t \in [t_k, t_{k+1})$

Mode Encoder:   $z \sim \mathcal{E}(X, \theta)$   $t = t_k$

Event Module:   $\psi_{z \to z'}, \ p_{z \to z'}$

| | |
|---|---|
| $z$ | latent mode (one–hot) |
| $X$ | collection of subtrajectories |
| $\{t_k\}$ | event times |
| $\theta$ | encoder parameters |
| $\omega$ | Neural ODE parameters |
| $F_z$ | mode–controlled Neural ODE |
| $\psi_{z \to z'}$ | jump networks |
| $p_{z \to z'}$ | normalizing flows |

## A.2   Gradient Pathologies

We provide some theoretical insights on the phenomenon of gradient pathologies with the simple example of a one–dimensional linear hybrid system with two modes and one *timed* jump,

$$
\begin{aligned}
\dot{x}_t &= \begin{cases} a x_t & t < \tau \\ b x_t & t >= \tau \end{cases} \qquad t \neq \tau \\
x_t^+ &= c x_t \qquad\qquad\qquad t = \tau
\end{aligned}
\tag{A.1}
$$

We let the system to evolve in a compact time domain $\mathbb{T} = [0, 1]$ such that $\tau \in \mathbb{T}$. Given an initial condition $x_0 \in \mathbb{R}$, the solution $x_1$ can be obtained as follows

$$
\begin{aligned}
x_\tau &= e^{a\tau} x_0 & &\text{1. integrate 1st flow until } t = \tau \\
x_\tau^+ &= c x_\tau = c e^{a\tau} x_0 & &\text{2. at } t = \tau \text{ apply jump} \\
x_1 &= e^{b(1-\tau)} x_\tau^+ = c e^{b(1-\tau)} e^{a\tau} x_0 = c e^{a\tau + b(1-\tau)} x_0 & &\text{3. integrate 2nd flow from } t = \tau \text{ to } t = 1
\end{aligned}
\tag{A.2}
$$

Alternatively, we can compactly write the solution at any time $t \in \mathbb{T}$ as

$$
x_t = \begin{cases} e^{at} x_0 & t < \tau \\ c e^{a\tau + b(t-\tau)} x_0 & t \geq \tau \end{cases}
\tag{A.3}
$$

Using the previous equation we can compute the gradient of solutions w.r.t. the parameters $a, \ b, \ c, \ \tau$. In particular, we have

$$
\begin{aligned}
\frac{\mathrm{d}x_t}{\mathrm{d}a} &= \begin{cases} t e^{at} x_0 & t < \tau \\ \tau k e^{a\tau + b(t-\tau)} x_0 & t > \tau \end{cases} \\
\frac{\mathrm{d}x_t}{\mathrm{d}b} &= \begin{cases} 0 & t < \tau \\ (t-\tau) k e^{a\tau + b(t-\tau)} x_0 & t > \tau \end{cases} \\
\frac{\mathrm{d}x_t}{\mathrm{d}c} &= \begin{cases} 0 & t < \tau \\ e^{a\tau + b(t-\tau)} x_0 & t > \tau \end{cases} \\
\frac{\mathrm{d}x_t}{\mathrm{d}\tau} &= \begin{cases} 0 & t < \tau \\ (a-b) k e^{a\tau + b(t-\tau)} x_0 & t > \tau \end{cases}
\end{aligned}
\tag{A.4}
$$

Now let us consider a loss function computed on the mesh solution points of the trajectory

$$
L = \sum_{k=1}^{K} \gamma(x_{t_s}), \qquad 0 < t_1 < \cdots < t_S < 1, \ t_s \neq \tau
$$

of which we wish to obtain the minimizers $a^*, \ b^*, \ c^*, \ \tau^*$ via e.g. application of gradient descent methods. The gradient of the cost function w.r.t. any of the parameters $\theta \in \{a, b, c, \tau\}$ is given by

$$
\frac{\mathrm{d}L}{\mathrm{d}\theta} = \sum_{k=1}^{S} \frac{\mathrm{d}\gamma(x_{t_s})}{\mathrm{d}x} \frac{\mathrm{d}x_{t_s}}{\mathrm{d}\theta}.
$$

Simultaneous estimation of both the optimal dynamic parameters $a^*, \ b^*, \ c^*$ and a randomly initialized event time $\tau^*$, will result in gradients of certain parameters to vanish or be completely incorrect.

Specifically, we note that parameter $\tau$ determines, beyond the specific time when the jump event occurs, also which parameters are responsible for computation of solution points $x_{t_s}$. Consider

the following two scenarios, where mode 1 is the first vector field of (A.1) and 2 is the second (post–event):

**(1)** Initialization of $\tau$ is an over–estimation of $\tau^*$ at the beginning of training. If this is the case, for $t_s$ such that $\tau > t_s > \tau^*$ the mode is missclassified i.e. should be 2, but is still 1 due to the delayed event time $\tau$. The gradient w.r.t $b$ of loss computed on solution points $x_{t_k}, \tau > t_s > \tau^*$ is then wrongfully set to zero.

**(2)** $\tau$ is an under–estimation of $\tau^*$. The same reasoning applies, except that for $\tau^* > t_s > \tau$ the mode is misslassified to 2, although it should be 1. This, in turn, affects the gradients for $b$, which results different than 0 despite the fact that $b$, from (A.1) should not be affecting the solution at points $t_s < \tau^*$. Any value taken by this gradient is thus incorrect, and can greatly affect the optimization procedure

We have shown how gradient pathologies exist even in the idealized linear case. In nonlinear systems with multiple events (including stochasticity) these effects can have a great empirical effect on a training procedure. The trajectory segmentation first approach of NHAs is designed to minimize their impact.

## A.3  Extensions and Limitations

**Automata reconstruction via symbolic regression**   NHAs with `categorical` encoders recover either a representation using the minimum number of modes necessary, corresponding to those of the system, or can be pruned due to immunity to mode mixing (discussed in Appendix B.3).

This property allows application of *symbolic regression* (SR) to reconstruct an analytic expression for each differential equation driving a system mode. This step grants domain experts a method to validate and certify the results, and enables construction of a human–readable automata representation for the SHS.

Clustered SR results can be improved by leveraging the *universal differential equation* approach employed in example by [10] for unimodal differential equations, by utilizing the decoder $F_z$ as an *interpolating* source of additional trajectory data for each mode.

**Latent hybrid automata from observations**   Learning methods for dynamical systems often introduce structure in latent space to enable control and identification from raw observations [41], [42]. Practical application for hybrid systems, such as robotic manipulation [43], locomotion [44], and traffic networks [2], might benefit from learning models structurally equipped with latent NHAs.

Optimal design of NHA modules for latent applications remains a difficult open question, as the analysis of deep models with latent spaces designed to evolve in continuous–time is also in its infancy.

**Unified benchmark for model development**   Despite the importance of hybrid systems in engineering, wide differences in techniques across domains have historically made it difficult to develop and preserve a unified set of benchmarks.

Evidence from other deep learning disciplines e.g. computer vision highlights the importance of consistent and curated benchmark datasets to track and measure the impact architecture and method optimizations. We argue further benchmark design, along with larger datasets, to be a necessary step required to trigger an `ImageNet`–like [45] moment for general neural differential equations and thus also NHAs. As an additional challenge, we note that performance of continuous neural models is in general highly impacted by the numerical method used for forward and backward inference, with optimal methods usually system or application dependent. This makes decoupling architecture improvements from the numerical underpinnings harder than for traditional models.

## A.4  Detailed Feature Comparisons with Related Work

Table 3 compares the proposed method with recent learning based approaches in terms of features. We use ✗ for features that are either absent or incompatible with a given method, or features that have not been tested or verified, although the method itself may be adapted to include it.

We consider the following:

- **Recovery of flows and events:** can state–space vector fields be learned along with the events. In [11], learned *latent* dynamics aid in the intensity parametrization of the point process. State–space dynamics are not learned simultaneously with point process maximum likelihood training. [13] trains a neural network vector field along with a parametrized event function.

- **Stochastic events:** has the method been shown to be compatible with stochastic events. The formulation of [13] can parametrize stochastic events via inverse sampling, but no experiments have been performed, likely due to difficulties in learning stochastic events from a full trajectory.

- **Mode identification:** does the method recover modes of a multi–mode hybrid system, and can the vector field approximate a different dynamics for each. NHAs are the first method to tackle this setting.

- **Adaptive end–time:** can the method adjust event times by calculating gradients with respect to integration end–times. Core contribution of [13] is an implicit differentiation formulation to adapt end-times. While adaptive segmentation has been discussed as being compatible with NHAs, no targeted experiments on this technique have been carried out. The extension is left as future work.

- **Intensity–free parametrization:** does the method use intensity–free parametrizations to avoid numerically solving integrals to sample from next–event densities. [11], [13] parametrize the intensity as only a single mode is considered. NHA use normalizing flows to approximate these densities directly, since intensity parametrization scale poorly as the number of system modes increases.

## A.5 Broader Impact

This work represents a first attempt in developing a data–driven, learning based technique for *stochastic hybrid system* (SHS) identification and control. As discussed in the main text, existing methods currently rely on strict assumptions that severely limit their utilization in practice. Applications domain the SHS formalism provides an accurate language to describe a target system are most likely to be affected by the availability of NHAs as a method to improve partial mathematical models using data or construct from scratch a model of the system and its automata representation. The impact of NHAs here is thus likely to be similar in scope to the impact of neural differential equations in science and engineering.

We do not anticipate significant negative environmental impact from the adoption of NHAs as these models are still orders of magnitude smaller than other large deep learning architectures for domains such as natural language.

## B  Experimental Details

**Hardware and software resources**  Experiments have been performed on a workstation equipped with a 48 threads AMD RYZEN THREADRIPPER 3960X, a NVIDIA GEFORCE RTX 3090 GPUs and two NVIDIA RTX A6000. All models and datasets fit in a single GPU. The software implementation of NHA leverages the `PyTorch` framework. ODE solvers and numerical methods for hybrid systems have been developed from scratch and are included in the submission.

**Common experimental settings**  In all experiments, unless specified, the NHA mode encoder $\mathcal{E}_\theta$ is capped with a `softmax` activation computing the probabilities of a categorical distribution from which then the one–hot mode $z$ is sampled.

| Method | Recovery of flow + events | Stochastic events | Mode identification | Adaptive end–time | Intensity–free parametrization |
|---|---|---|---|---|---|
| NJSDE [11] | ✗ | ✓ | ✗ | ✓ | ✗ |
| Neural Event ODE [13] | ✓ | ✗ | ✗ | ✓ | ✗ |
| **NHA** (this work) | ✓ | ✓ | ✓ | ✗ | ✓ |

**Table 3:** Feature comparison between neural models for hybrid systems. ✗ is used to indicate features that are either not compatible, or have not been verified in the original work.

| Simulation Hyperparameter | Value |
| --- | --- |
| Number trajectories | 40 |
| ODE solver | Dormand-Prince |
| Tolerances (abs, rel, event) | $10^{-6}, 10^{-6}, 10^{-4}$ |
| $\tau_{\text{off}}$ | 3 |
| $\eta$ | 1 |
| $n_{\text{ack}}$ | 2 |
| $p_{drop}$ | 0.05 |
| $\kappa$ | 4 |

| Training Hyperparameter | Value |
| --- | --- |
| Training iterations (mode recovery) | 8000 |
| Encoder $\mathcal{E}_\theta$ learning rate | $3 \cdot 10^{-4}$ |
| Decoder $F_z$ learning rate | $10^{-2}$ |
| Optimizer | AdamW |
| $\mathcal{E}_\theta$ layer dimensions | $[\cdot, 64, 65, 64, m]$ |
| $\mathcal{E}_\theta$ activation | ReLU |
| $\mathcal{E}_\theta$ dropout | $[0.3, 0.3, 0.3]$ |
| $F_z$ layer dimensions | $[2 + m, 2]$ |
| Training iterations (event training) | 4000 |
| Learning rate | $2 \cdot 10^{-3}$ |
| Optimizer | AdamW |
| $\psi_{z \to z'}$ layer dimensions | $[2, 32, 2]$ |

**Table 4:** Hyperparameters of **[Left]** TCP data simulation **[Right]** NHA in the TCP experiment, mode recovery and event module training.

Gradients through the sampling operation are computed via *straight–through–estimation* (STE) [27], which can be implemented with a `stop_gradient` (e.g. `detach` in `PyTorch`) operation present in modern deep learning frameworks. Let $z$ be the one–hot representation of system mode, and $p$ a vector of probabilities for the corresponding `categorical` distribution, computed as output of a neural network parametrized NHA encoder $\mathcal{E}_\theta$. STE can be realized in a single line as $z - p.\texttt{stop\_gradient}() + p$. STE ensures the output of the encoder $\mathcal{E}_\theta$ is strictly one–hot encoded, while simultaneously ensuring that the gradients backpropagate directly through the probabilities.

The data–controlled Neural ODE decoder $F_z$ incorporates mode information to select

$$\dot{x} = \sum_{i=1}^{m} z_i \, f_i(t, x, \omega_i)$$

where $z$ is one–hot encoded, and thus only a single neural network vector field $f_i$ with parameters $\omega_i$ determines the solution.

### B.1 Identification of Reno TCP

**Experimental setup** Tables 4 provide hyperparameters for data simulation and training of NHA. Unless otherwise specified simulation, training and testing is repeated for 10 different random seeds.

The complete experiment on learning the Reno TCP system involves multiple stages: (i) mode recovery and (ii) training of the NHA event module. As baselines for (i), we consider several Neural ODE variants similar to NHA decoder $F_z$, with latent $z$ obtained in different ways. Latent Neural ODEs obtain $z$ as sampled for Normal distribution $\mathcal{N}_\theta := \mathcal{N}(\mu_\theta, \sigma_\theta)$ parametrized by a neural network matching the architecture of $\mathcal{E}_\theta$. Reparametrization is used to backpropagate through the sampling procedure. *Data–controlled Neural ODEs* (DC–NODEs) are comparable to Latent Neural ODEs model with the major difference in the computation of latents as $z = g_\theta(X)$ with $g_\theta$ once again matching $\mathcal{E}_\theta$. All decoders $F_z$ are equivalent, except in the case of Augmented Neural ODEs (the Neural ODEs is zero–augmented [9] i.e. $z := 0$) where the absence of the encoder is balanced by a more expressive decoder with three–layers. The normalizing flows $\psi_{z \to z'}$ in NHA event modules are designed as spline flows [32] with two layers. We use the standard implementation of spline flows in `Pyro` [46].

During mode recovery, all models are trained on 5–folds of 5. Training is performed by parallel integration across subtrajectories $X_i$ using the `Runge-Kutta`4 explicit solver. All gradients are computed via reverse–mode automatic differentiation. We test models with lowest cross–validation reconstruction MSE loss, since cross–validation v–score would not be available without ground–truth labels. We find that lowest reconstruction loss often correlated with best v–measure. Baselines for mode clustering (ii) include standard clustering algorithms k–means++ [29], hierarchical [30] and DBSCAN [31]. We use `scikit-learn` [47] implementation of all baseline algorithms. We perform light hyperparameter tuning on ground–truth labels for k–means++ and hierarchical to optimize their

performance in the range $m \in [3, 5, 10]$. DBSCAN is similarly tuned to optimize its performance with parameter $\epsilon$ indicating the maximum size of neighourhoods around a data point. Subtrajectories classified as noise by DBSCAN are counted as incorrectly clustered.

We observe DBSCAN performance to be correlated to NHA self–supervised mode recovery. Both methods excel when density of data points under some metric is indicative of cluster separation. However, NHA self–supervision relies on the additional inductive bias of data points in a subtrajectory representing observations of a solution of an ODE, whereas DBSCAN does not. The denser the trajectories, the more restricting the Neural ODE representation limitations, and the easier each cluster is to find. Due to the similarity in their working principle, DBSCAN performance can be used as a quick sanity check to determine whether the dataset is suitable for NHA mode recovery.

## B.2 Robustness to Segmentation Noise

We investigate robustness of NHA mode recovery to a noisy segmentation in subtrajectories $X_i$. To simulate incorrect segmentations, we collect segmentation indices and perturb them by adding or removing an uniformly sampled from $[1, 10]$. Each index has a $p$ probability of being corrupted by noise, and we repeat mode recovery with $p \in [0.1, 0.3, 0.5]$ (3 times per $p$). Shifting left or right by values sampled from $[1, 10]$ results in significant data corruption; certain subtrajectories, being shorter than 10 points, can be completely absorbed into a different subtrajectory. Table 5 reports the differences in v–measure compared to the results of Section 5.1.

|  | $\Delta$ v–score | | |
|---|---|---|---|
| **Model** | $p = 0.1$ | $p = 0.3$ | $p = 0.5$ |
| NHA–3 | $-0.10$ | $-0.27$ | $-0.38$ |
| NHA–5 | $-0.09$ | $-0.24$ | $-0.34$ |
| NHA–10 | $-0.07$ | $-0.25$ | $-0.33$ |

**Table 5:** self–supervised mode recovery v–measure performance loss due to noisy segmentation of the TCP dataset discussed in the main text. We evaluate under an increasing degree of data corruption. Hyperparameter $p$ indicates the probability for a subtrajectory $X_i$ to be subject to a noisy segmentation i.e. to have the index determining its initial condition be perturbed and shifted either left (before) or right (after).

## B.3 Switching Linear System and Mode Mixing

**Experimental setup** We considered the two–dimensional switching linear system reported in [20], described by the dynamics

$$(\dot{x}_t, \dot{y}_t) = f(x_t, y_t) := \begin{cases} (-y_t, x_t + 2) & \text{if } x_t \geq 2 \\ (-1, -1) & \text{if } x_t < 2 \wedge y_t \geq 0 \\ (1, -1) & \text{if } x_t < 2 \wedge y_t < 0 \end{cases} \tag{B.1}$$

We performed an ablation study on the effect of the categorical sampling for the mode selection in NHAs in presence of redundant "free" modes. In particular, we considered the following learning model

$$(\dot{x}_t, \dot{y}_t) = \sum_{i=1}^{4} w_t^i f_i(x_t, y_t) \tag{B.2}$$

with one redundant mode. $F_i$ $(i = 1, 2, 3, 4)$ was two-layers neural networks with 32 neurons each, `softplus` activation on the first hidden layer and hyperbolic tangent activation on the second one. We then defined two variants of the model: a first variant with $w_t = (w_t^1, w_t^2, w_t^3, w_t^4)$ directly obtained via `softmax` normalization of the output of a neural network $g$,

$$w_t = \texttt{softmax}\, g(x_t, y_t);$$

and a second one where $w_t$ is obtained by a categorical sample conditioned by $g(x_t, y_t)$, i.e.

$$\forall t \quad w_t \sim \texttt{categorical}(\texttt{softmax}\, g(x_t, y_t))$$

$g$ was fixed as a neural network made up by two layers with 64 units and `SiLU` (`swish`) activation. The two models were trained on a $L_1$ reconstruction loss of nominal trajectories of the system (B.1). We introduced a regularization term penalizing the squared error on un–normalized finite differences of nominal/reconstructed trajectories as a proxy for the vector field information.

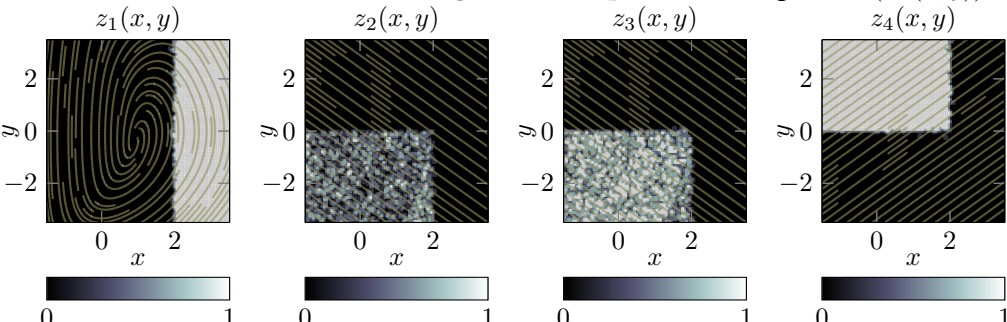

**Figure 10:** Reconstructed conditional vector fields $F_z$ and corresponding mode classification boundaries. The `categorical` encoder uses two identical modes for a single ground–truth vector fields.

**Mode pruning** Uniqueness theorems for ODE solutions guarantee that, given an initial condition and a mode latent code $z$, the decoder $F_z$ will always produce the same trajectory. Immunity to mixing for categorical latents enables mode pruning and recovery of a minimal representation. If $\mathcal{L}_r$ saturates, the encoder has not been initialized with a sufficient number of modes $m$. Redundant modes may be pruned, in example, by merging them if a similarity measure between the corresponding vector fields $F_{z_i}$, $F_{z_j}$ e.g. difference in a given norm calculated on data trajectories, is *small enough*. Figure 10 provides an example result of the second scenario discussed in Section 5.2, where `categorical` NHA encoders use more than a single latent mode for a target underlying mode. However, due to their immunity to mode mixing, the vector fields are equivalent, and can be merged. We show this in Figure 9, where the L1–norm between $F_2$ and $F_3$ is shown to be small in the region where the corresponding modes 2 and 3 are active.

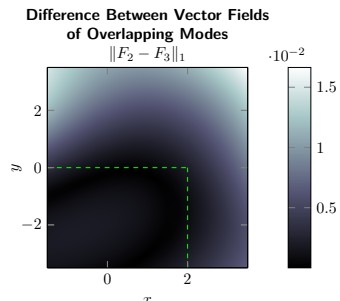

**Figure 9:** Similarity between learned mode vector fields $F_2$ and $F_3$ of Figure 10. The two vector fields are equivalent in the region of interest, as indicated by the L1–norm, and the corresponding modes can thus be merged.

### B.4 End–To–End Learning of Hierarchical Controllers for Dynamical Systems

**Experimental setup** Our objective is to control a swarm of differential drive robots moving on a planar space. The system dynamics are

$$
\begin{aligned}
\dot{x}_1 &= u_v \cos\theta \\
\dot{x}_2 &= u_v \sin\theta \\
\dot{\vartheta} &= u_r
\end{aligned}
\tag{B.3}
$$

where $\vartheta$ is the orientation of a single robot. The control $[u_v, u_r] \in \mathbb{R}^2$ is obtained via the low–level, time–invariant feedback controller $u_z := u(x_1, x_2, \vartheta, z)$, with $z$ produced and switched every 5 seconds by the planner. Both the controller and the policy $\pi$, are parametrized by a neural network. We use `Adam` for both networks, with learning rates $10^{-3}, 10^{-3}$.

The training of low–level controller $u_z$ and high–level planner $\pi$ is carried out concurrently. We perform batch training on robot swarms of $N = 80000$. At the beginning of each episode, we sample the initial conditions uniformly in a square region, with each robot rotated according to a random orientation also uniformly sampled in $[0, 2\pi]$. During each episode, we simulate the switching and control behaviour of each robot with respect to two pre-defined map layouts, both shown in the alternating resource pattern of the upper-left plot of Figure 8a. Each map layout has $M = 5$ resource locations, where two auxiliary scalar variables $r_1$ and $r_2$ specify, for each resource, its planar location.

We train policy networks $\pi(x_1, x_2, r_{11}, r_{21}, ..., r_{1m}, r_{2m})$ where the inputs correspond to the concatenated robot states, together with the flattened resource locations. The $M-$dimensional softmax

output determines a categorical probability distribution over the resources is then used to sample a resource target.

A reward can be assigned to each robot based on the ability to select the correct target. For each robot, we generate a reward of 1 if the selected target is correct, and 0 otherwise. Let $G(s_i, p_i)$ be the reward of robot $i$ in the swarm, we can compute the reward for a swarm in a map by $R = \sum_i^N \frac{G(s_i,p_i)}{1000}$ where $s$ is the categorical sample from the distribution over the targets, and $p$ are the reference targets computed by $\arg\min_j ||[r_{1j}, r_{2j}] - [x_1, x_2]||$ (note $R \in [0, 80]$ for any one episode), with $x_1, x_2$ being the robot location at the time of switching. The target selector is trained by minimizing a $\mathcal{L}_\pi = -\frac{1}{T} \sum_t^T ln(\pi(s_p^{(t)}) * R)$, where T is the number of alternating maps, and $s_p^{(t)}$ denotes the concatenated robot location and resource map with respect to the $t^{th}$ map layout used for training.

The target selected by the planned informs low–level controller $u_z$ via the corresponding resource location. In particular, we provide as input to $u_z$ the coordinates $z := [r_1, r_2]$ of the target chosen by $\pi$. This augmented state is used by $u_z$ to resolve the robot's dynamics and drive the swarm closest to their selected targets. We train $u_z$ by solving a continuous–time optimal control problem with a terminal RMSE loss between the state reached by the robot and the objective set by the policy planner. Here, we integrate the system using the adaptive–step `DormandPrince` [48] solver with tolerances $10^{-3}, 10^{-3}$.

**Discussion of results**   Figure 8 shows the average reward and control loss of the robot swarm during training, with both trends converging after 4000 episodes. Figures 8a and 8b show the generated control of a randomly generated swarm of 100 robots on two new maps. In the first map, the targets consist on alternating patterns of the learned map layouts, generating a straight line pattern which correctly captures the greedy robot policy imposed. The second map consists instead of a new, unseen, map layout within the alternation. The trained model is capable of generalizing the planning and control strategy to account for the new map layout, by redirecting the robots onto their closest resource in a wave-like pattern. On the first tested map, the model achieves $99.8\% \pm 0.8\%$ average target accuracy for the 100 robot tested batch. On the second tested map, the model achieves $98.5\% \pm 1.95\%$ average target accuracy for the 100 robot tested batch.

## C   Realization of NHAs

### C.1   Software Implementation of Hybrid Integration

We provide documented `Python` pseudo–code for the hybrid system adaptive integration algorithm used for dataset generation. This function can handle hybrid systems with multiple modes and transitions. Each possible event requires its own callback function with `check_event` and `jump_map` methods. We provide an example of one such callback under `odeint_hybrid`.

```python
def odeint_hybrid(vf, x, t_span, solver, callbacks, atol, rtol, event_tol):
    """ODE solver for hybrid systems with multiple events."""
    # initialize event state tracker, one boolean for each possible event
    # (or edge in the automata representation of the SHS).
    event_states = [False for _ in range(len(callbacks))]
    dt = initial_step_size(f, k1, x, t, solver.order, atol, rtol)

    while t < t_span[-1]:
        # tentative step
        x_step, x_err = solver.step(vf, x, t, dt)

        # check whether any event
        # has been triggered in the interval [t, t + dt]
        new_event_states = [cb.check_event(t + dt, x_step)
                                    for cb in callbacks]

        # has any event state moved from `False' to `True' in [t, t + dt]?
            triggered_events = sum([(zp != z) & (z == False)
                    for zp, z in zip(new_event_states, event_states)])
            # if an event / mode transition has been triggered,
```

```
21              # find exact event time and state
22          if triggered_events > 0:
23              x, t = root_find_event(max_iters, event_tol)
24
25                  # if there is a conflict and multiple events are triggered,
26                  # takes always the one with smaller ID
27                  zp = min([i for i, ev in enumerate(new_event_states)
28                           if ev == True])
29
30              t = t + dt
31              # save state and time BEFORE and AFTER jump
32              sol.append(x)
33              eval_times.append(t)
34
35                  # apply jump func.
36                  x = callbacks[zp].jump_map(t, x)
37
38              sol.append(x)
39              eval_times.append(t)
40
41          # when there are no events,
42          # proceed as usual with adaptive integration
43          else:
44              error_ratio = compute_error(x_step, x_err, atol, rtol)
45              accept_step = error_ratio <= 1
46
47              if accept_step:
48                  t = t + dt
49                  sol.append(x)
50                  eval_times.append(t)
51
52              else:
53                  dt = adapt_step(dt, error_ratio, safety,
54                                  min_factor, max_factor, order)
55
56      return eval_times, sol
```

The callbacks are in the form:

```
1  class StochasticEventCallback(nn.Module):
2
3          super().__init__()
4          self.exponential = Exponential(1)
5
6      def initialize(self, x0):
7          # sample one `s' for each batch in x0, to identify events
8          # as described in Section 2. Exponential, instead of Uniform
9          # is used to avoid `log' computations.
10          # Should be sampled again after every event is triggered.
11          self.s = self.exponential.sample(x0.shape[:1])
12
13      def check_event(self, t, x):
14          raise NotImplementedError
15
16      def jump_map(self, t, x):
17          raise NotImplementedError
```