# OpenReview forum: "Neural Hybrid Automata: Learning Dynamics With Multiple Modes and Stochastic Transitions"
_NeurIPS.cc/2021/Conference — NeurIPS 2021 Poster_

### Official Review · Reviewer_kV6b · 2021-07-16

**Rating:** 6
**Confidence:** 2

**Summary:**

This paper introduces a complete pipeline for controlling with multi-modal transition and event handling. The method shown in the paper is based on normalizing flow and neural differential equations. In the experiments, the paper shows the efficacy in a few controlling cases. The method first segments the trajectory into multiple partitions, then performs the reconstruction using normalizing flow.

**Limitations And Societal Impact:**

Yes

**Main Review:**

The problem formulation and presentation are fine. The figures (especially figure 2) are not really informative enough. The figures and captions should deliver straightforward ideas without ambiguities. The arrows are lack notations or explanations. It will be good to improve them.

The cross-reference in lines 254 and 259 are broken. The citation format is not uniform. They are not technical problems but it will be helpful to fix them.

My main reservation is the robustness of the proposed method. Does it perform constantly well against noises? A few quantitative results will help the reader understanding it.

**Time Spent Reviewing:**

4

---

> ### Author Response · Authors · 2021-08-10
> **Response (kV6b)**
>
> We thank the reviewer for their valuable feedback and positive comments about our work.
>
> 1. We note that the supplementary material at the time of submission contains a quantitative evaluation (**Section B2**) on robustness to noise of the proposed *Neural Hybrid Automata* (NHA). In particular, we measure performance of NHA and clustering baselines on TCP data subject to noise injected during the segmentation of trajectories into sub-trajectories. The mode clusters recovered by NHAs are more robust than baselines such as DBSCAN. The experiment further considers increasing noise magnitudes. In example, under fairly severe noise level p=0.3 the v-measure for  DBSCAN, an existing state-of-the-art for trajectory clustering method, drops ~ 0.3 lower than NHA-10. We thus conclude that NHA is sturdy to noise injection, especially with respect to other trajectory clustering algorithms. We hope this addresses the question about robustness and we'd be glad to further elaborate on this aspect.
>
>
> 2. We have fixed the cross-reference typo and double-checked citation style consistency. Figure 2 provides a high-level overview of the entire NHA procedure. The arrows signify a temporal dependence, as steps of the procedure outlined in the diagram occurs in a specific order. In particular, each block is described in the text: **Flow Segmentation** corresponds to the introduction of **Section 4**, **Flow and Mode Self-Supervision** to the first paragraph of **Section 4** and **Train Event and Jump** to the second. We have improved the description of our figures across the manuscript and made the connections to each paragraph clearer. We also note that the supplementary provides an additional summary of each NHA module to guide the reader (Section A1 has a notation table and a textual description).
>
> **Conclusions:**
> We remain available to address any further concern and incorporate additional feedback on style and figures. We would also be glad to elaborate on aspects of the proposed method in the hope of increasing reviewer confidence in the work and score.

---

### Official Review · Reviewer_KEKy · 2021-07-16

**Rating:** 7
**Confidence:** 3

**Summary:**

This paper considers a modeling class (NHA) based on stochastic hybrid systems that can represent continuous-time processes with discrete events. They discuss the foundations of these models of the dynamics model, discrete latent selector, and event module to select events, and proposed a self-supervised mode recovery and supervision for events and jumps. NHA excel in diverse experimental settings on Reno TCP, mode mixing in switching systems, and behavioral control of wheeled robots.

**Limitations And Societal Impact:**

The authors discuss some limitations and insights in S5.4 but otherwise did not have an extensive discussion in this space.

**Main Review:**

I like the paper! The foundations are reasonable and the experiments are well-investigated, analyzed, and documented for the community.
S5.4 provides a nice overview of their insights and observations, and Fig 7 shows a nice visualization of the learned mode classification boundaries.

My biggest question about this paper is on the experimental comparisons to neural ODE-based methods --- could NHAs be applied in many of the same settings where NODEs are already being used? The experiments presented in the paper have the feeling of new tasks that haven't been as considered before, which are interesting for the community to think about, but it would also be insightful if these methods could improve upon NODE-based modeling in the existing settings too.

**Time Spent Reviewing:**

1 hour

---

> ### Author Response · Authors · 2021-08-10
> **Response (KEKY)**
>
> We thank the reviewer for the positive feedback.
>
> **NHAs in standard Neural ODE (NODE) settings:**
>
> NHAs can be applied whenever a neural ODE can. Our *Neural Hybrid Automata* (NHA) procedure can be thought of as a generalization that can handle systems with many modes of operation and events of different types, including mode-shifting and not-mode-shifting. Vanilla Neural ODEs are typically applied to domains that are a priori known to be unimodal and event-free, where the NHA dynamics module can also be equivalently used. However, a number of real-world trajectory datasets are generated by systems with an unknown number of modes or events. Here, NHAs provide a more flexible model prior in settings with uncertainty about the nature of the underlying process.
>
> We also note that jumps can be used to address expressivity limitation of vanilla Neural ODEs, by allowing in example to tackle the reflection map benchmark presented in [1,2].
>
> **Conclusions:**
>
> We would be glad to answer any further questions. We are always available to greatly expand upon specific aspects of the method, particularly if it can help improve further improve reviewer confidence.
>
> [1] Augmented Neural ODEs, Dupont et al.
>
> [2] Dissecting Neural ODEs, Massaroli et al.

---

### Official Review · Reviewer_hhU5 · 2021-07-17

**Rating:** 6
**Confidence:** 3

**Summary:**

This paper proposes a solution called 'Neural Hybrid Automata' to model real world dynamical systems with multiple modes of operation and stochastic switching between modes. Such problems are plentiful in a variety of engineering domains. The authors address this using three modules: Dynamics model, Latent model to select the mode and Event model to predict the scitching.

**Limitations And Societal Impact:**

Some limitations have already been mentioned in the paper. Please see the main review for additional ones.

**Main Review:**

This paper relies heavily on recent work around the use of Neuro-ODE, Hamiltonian and Lagrangian dynamics. From that perspective this is quite timely and relevant. The maths look good to me. I consider this very interesting, but there are a couple of things that can be improved.

- The first 2 modules of the work are quite solid. However, the last module that focuses on the switching is not as convincing, It is unclear how well it can capture stochasticity for example in a stock price as it reacts to market moving events. This is an area that has been well explored and there exists plentiful literature. Since not enough experiments have been conducted, it is unclear how this extends the state of the are as far as usefulness goes.

- Related to the above, the paper doesn't conduct enough experimentation for real world problems and that in my mind is a limitation. The experiments that are shown with synthetic data are not convincing enough.

- I would also like to see how it compares against known state of the art solutions.

**Time Spent Reviewing:**

3 hours

---

> ### Author Response · Authors · 2021-08-10
> **Response (hhU5)**
>
> We thank the reviewer for their feedback and positive comments. We are glad the reviewer found the paper to be interesting and relevant.
>
> **Event module and stochasticity:**
>
> The event modules of NHAs are designed to handle stochasticity in event times, not stochasticity in the continuous-time dynamics $F_z$. We assume the dynamics to be deterministic, with the only stochasticity present in the event time and mode transitions. In other words, as standard in temporal point processes, we model event times $t_k$ (or equivalently inter-event times) as random variables. The densities of these random variables are learned from data as described in lines 155-160 of Section 4. We have clarified this point in the text.
>
> To expand on the example provided by the reviewer, stock price data do not satisfy the assumption of deterministic dynamics, and are thus outside the scope of this work. We note, however, that extensions to SHSs with stochastic dynamics are possible, and that replacing deterministic Neural ODEs with a *Stochastic Differential Equation* primitive (i.e. Neural SDE [1]) in NHA dynamics, along with appropriate assumptions on the class of events, could in principle be feasible. This extension would allow to combine and learn multi-mode Brownian dynamics with stochastic jumps, which certainly is compatible with a variety of applications in finance including market regime switching detection. Overall, we agree that finance is a fruitful domain in which to perform follow-up work in. This discussion and potential future work direction have been added in the manuscript.
>
>
> **Experiments and state-of-the-art:**
>
> A machine learning framework for hybrid systems and SHS identification is new. To the best of our knowledge no holistic data-driven procedure existed before the proposed NHA, including dynamics, event and mode learning. As outlined in Section 2.3 and 6, the closest attempts from the controls and machine learning community such as [2,3] fall short by not considering aspects such as multi-modality or event stochasticity.
>
> For these reasons, we select and consider state-of-the-art (SOTA) baselines corresponding to **each module** of the *Neural Hybrid Automata* (NHA). These comparisons include many standard baselines that are reasonable indicators of SOTA performance in their respective task, such as Latent Neural ODEs for flow reconstruction and DBSCAN for trajectory clustering. We would be happy to address further questions, especially if the reviewer can point to important related work or methodology for SHSs missing from the current version of the manuscript.
>
> Due to the lack of prior work on learning SHSs from data and related benchmarks, we had to design a novel, dedicated set of evaluations:
>
> 1. **Experiment 1 -- *Learning Transmission Control Protocol* dynamics (TCP):** the TCP Reno model used is one that has been extensively studied in the literature [4] and guides theoretical exploration on design of optimal TCPs. While there are learning approaches to the estimation of certain metrics of such controllers [5], NHA provide a first novel solution to the challenging problem of TCP identification without a priori knowledge of the number of modes. Solving this problem has a variety of practical, real-world applications. For example, NHA models could be in principle used to optimize parameters of a TCP scheme.
> 2. **Experiment 2 -- Mode-mixing:** is designed to specifically address limitations of prior work [2]. Here, we highlight the existence of mode-mixing, a novel issue in learning hybrid system, and verify that straight-through-sampling addresses it.
> 3. **Experiment 3 -- Hierarchical robot control:** verifies the out-of-the-box applicability of NHAs to a different domain, target-aware control. Scaling to real-world systems here would require adapting the model used for NHA dynamics, while keeping the other components as described in our work. We believe hierarchical control is a domain with various direct real-world applications, and we provide a complete blueprint by showing how to conceptually leverage the NHA framework for such tasks.
>
>
> **Conclusions:**
>
> We gladly welcome further comments and questions by the reviewer, and hope the current discussion addresses all technical concerns raised.
>
> [1] Scalable Gradients for Stochastic Differential Equations, X. Li et al.
>
> [2] Learning Neural Event Functions for Ordinary Differential Equations, R.T. Chen et al.
>
> [3] Learning Symbolic Representations of Hybrid Dynamical Systems,  Daniel L. Ly et al.
>
> [4] Stochastic Hybrid Systems: Application to communication networks, J.P. Hespanha et al.
>
> [5] A Deep Learning Approach to Dynamic Passive RTT Prediction Model for TCP, Hagos et al.

---

> ### Comment · Reviewer_hhU5 · 2021-08-13
> **Post Rebuttal**
>
> I appreciate the author's attempts to improve the paper, so I would like to increase my score to 6.

---

> > ### Author Response · Authors · 2021-08-14
> > **Response II (hhU5)**
> >
> > We are glad the reviewer found the response meaningful and chose to increase the score to 6. We noticed that the score has not been changed in the official review, and kindly ask the reviewer to check and edit the review to match the comment.
> >
> > We remain available to answer further questions if needed.

---

### Decision · Program_Chairs · 2021-09-27

**Decision:**

Accept (Poster)

**Comment:**

After clarifications made by the authors in their rebuttal, all reviewers agreed about the merits of this work, and recommended acceptance. In the final version, please incorporate the feedback given in the reviews, and add the important clarifications of the rebuttal.